# Achieving High-Precision, Low-Cost Microfluidic Chip Fabrication with Flexible PCB Technology

**DOI:** 10.3390/mi15040425

**Published:** 2024-03-22

**Authors:** Andres Vanhooydonck, Thalissa Caers, Marc Parrilla, Peter Delputte, Regan Watts

**Affiliations:** 1Faculty of Design Sciences, Department of Product Development, University of Antwerp, Paardenmarkt 94, 2000 Antwerp, Belgium; regan.watts@uantwerpen.be; 2Laboratory for Microbiology, Parasitology and Hygiene (LMPH), Faculty of Pharmaceutical, Biomedical and Veterinary Sciences, University of Antwerp, 2610 Antwerp, Belgium; thalissa.caers@hotmail.com (T.C.); peter.delputte@uantwerpen.be (P.D.); 3A-Sense Lab, University of Antwerp, Groenenborgerlaan 171, 2010 Antwerp, Belgium; marc.parrillapons@uantwerpen.be

**Keywords:** microfluidics, PCB, lithography, PDMS, prototyping, biocompatibility, POCT, LOAC, microfabrication

## Abstract

Soft lithography has long remained the state of the art to generate the necessary micropatterning for molded microfluidic (MF) chips. Previous attempts to use printed circuit boards (PCBs) as a cheap and accessible alternative to expensive lithographed molds for the production of PDMS MF chip prototypes have shown their limitations. A more in-depth exploration of using PCBs as a mold substrate and a novel methodology of using flexible PCBs to produce highly accurate MF chips is reported here for the first time. Cross sections highlight the improved accuracy of this method, and peel testing is performed to demonstrate suitable adhesion between the glass substrate and PDMS cast. Positive cell growth viability showcases this novel method as a high-accuracy, high-accessibility, low-cost prototyping method for microfluidic chips while still maintaining all favorable properties provided by the PDMS material.

## 1. Introduction

Soft lithography (SL) has remained the state of the art to produce microfluidic (MF) chips for over 30 years [1,2]. To enable the faster development of MF platforms, prototyping is essential to validate and test designs in a rapid and iterative way. By using this approach, the final product is developed more quickly, having eliminated non-viable ensembles in the iteration phase [3]. One field that greatly benefits from MFs is point-of-care testing (POCT) [4]. A POCT device is a portable medical device that can be used to conduct a specific assay near the patient as an alternative for central laboratory testing. This provides faster decision making and leads to better healthcare outcomes [5]. The prototyping and validation of lab-on-a-chip (LOAC) devices are crucial steps towards miniaturizing laboratory tests into integrated POCT solutions in the diagnostic healthcare market segment.

Polydimethylsiloxane (PDMS) has long been the standard material for casting MF chips because of its favorable properties. It is optically transparent, can bind to glass, is gas permeable, inert, sterilizable, inexpensive, and patent-free [3,6,7]. To make a functional glass–PDMS chip, first, a mold is created. Due to the high detail that soft lithography offers as a mold micropatterning process with accuracy up to 500 nm [8] and the highly desirable properties of PDMS, this combination has remained the standard for an exceptionally long time [2].

While lithography is widely regarded as a cost-effective technique within the field of microfluidics, its relative expense becomes evident when compared to PCB-based master molding or conventional rapid prototyping methods such as various 3D printing methods. The higher costs associated with lithography stem from the use of expensive silicon wafers, sophisticated machinery, stringent clean room requirements (ISO4-ISO5) [9,10], and lower production volumes. In contrast, PCB production benefits from the use of inexpensive, readily available materials, lower cleanroom standards (ISO 7) [11], and from the economies of scale associated with high-volume production [12]. This stark contrast in cost structures makes PCB-based approaches significantly more accessible for rapid prototyping, offering a more cost-efficient solution that better supports iterative design processes. To make MF chip processing more accessible, we propose a fast and cheap proto-typing method for mold micropatterning based on PCBs to replace the lithographed master mold.

PCBs have been used frequently in the field of MF [13,14,15]. The holy grail of this field is considered a completely integrated MF system called the µTAS [16,17]. Very often, the reagents in these MF systems are manipulated or measured with electronic systems such as heating elements, electromagnets, spectrometers, and potentiometers, among others [13]. Therefore, the integration of PCBs with MF is common in the literature. Even though the integration of PCBs within MF platforms is commonplace, and given that PCBs have very mature and reliable production methods comparable to SL, including micropatterning through chemical etching, they have rarely been used as a mold.

Previous research has already shown that it is possible to use PCBs as a molding substrate for microfluidics [18,19]. Tiwari et al. described a process of do-it-yourself (DIY) etching of copper-clad laminate (CCL) to create molds used for casting MF chips while requiring minimal infrastructure [18]. One limitation, however, is the roughness of the substrate, transferred to the cast surface which did not allow the PDMS to properly bond to glass but only to another PDMS surface, resulting in flexible MF chips that can be easily deformed and may not be able to withstand the pressures and forces that are involved in many microfluidic applications. Flexible MF chips are also not able to retain the same tolerances as rigid PCBs. Tiwari et al. overcame this by using PDMS-coated glass slides before trying to bond them to the PDMS from the PCB. The in-house production also requires trained personnel and manual labor which is costly. In other research, the possibility of using commercial, commoditized PCB services to outsource the manufacturing process of the PCB-mold led to promising results [19]. However, the same roughness issue remained and led to poor adhesion between the PDMS cast and glass after plasma bonding. They tried to overcome this by polishing and cleaning the PCB thoroughly [20] or covering the PCB in epoxy resin before casting [21].

Herein, we propose a novel method using a PCB, covered in a solder resist to create a smoother surface finish suitable for bonding directly to glass. We also propose another novel method of using flexible PCBs as an alternative master mold. By using these PCBs as master molds, the final chip can be produced quickly and easily while maintaining all the favorable properties of using a PDMS–glass combination for the MF chip. Ordering these molds made by a mature manufacturing process allows for more accurate and repeatable results by excluding errors that could arise from the manual mold post-processing steps. Flexible PCBs prove to have many advantages compared to previously proposed fiberglass-reinforced epoxy laminated, or FR-4, PCBs. This is demonstrated in this work by comprehensive peel testing, functionality testing, analysis based on optical and scanning electron microscopy (SEM), simple surface modification, and cell growth viability testing using a resazurin-based assay test on retinal pigment epithelial cells (ARPE-19).

## 2. Materials and Methods

### 2.1. PCB Types and Configurations

In this paper we will discuss two types of PCBs; the ubiquitously used rigid FR-4 PCB made from FR-4 copper clad laminate (CCL) and more accurate flexible circuit boards (FCB) with a flexible polyimide (PI) core. The laminates used for both types of PCBs comprise either a FR-4 fiberglass or a PI core, with a copper foil adhered to one or both sides. The thickness of this copper layer, important for shaping the eventual track heights, is conventionally measured in ounces per square foot (oz/ft^2^), typically ranging from 0.25 oz (equivalent to 9 µm) up to 3 oz (105 µm), and can extend up to 20 oz (700 µm) for certain extreme applications.

The boards will be used only for the micropatterning of fluidic channels and therefore do not have to follow the design rules required for electrically functional circuit boards.

When ordering a custom PCB, one can choose from many standard features including copper tracks, surface-mounted pad designs, ground planes, solder resist layer, screen-printing layers, vias or through-hole plating, among others elements, but not all features are desirable for generating molding geometries (for example, multi-layer PCBs or hole and slot placement in the board substrate). The most relevant features are discussed.

After some initial experimentation, several elements were identified to be tested for microfluidic patterning on 5 configurations of rigid FR-4 boards and flexible boards:

(i) Rigid PCB with only the exposed base substrate surrounding the tracks covered in the solder resist, with optionally tinned copper tracks (Figure 1a). This method provides sharp copper tracks combined with a smooth solder resist-covered FR-4 base substrate surrounding it. However, the solder mask layer is applied in a separate step after the copper tracks are formed, typically using a screen-printing method. By adding this second operation, the alignment of said layers can never be perfect, so a small gap is always present between the tinned tracks and the solder resist layer in the design to avoid overlap. When this gap is removed in the design drawings, the solder mask layer can overlap with the tracks. This gap between the tracks, however, results in a protruding geometry, making it impossible to plasma-bond the PDMS cast made with this mold to glass.

(ii) Rigid PCB with exposed copper tracks with or without tinning, no solder resist (Figure 1b). Bare copper tracks have been used in past research [18,19] and result in sharp track outlines with a rectangular cross section. The track produces favorable track geometry. However, due to the production technique of CCL boards, the FR-4 base substrate is roughened to ensure proper adhesion to the copper layer during the production of this laminate material. This is especially crucial for the adhesion of smaller tracks after etching [22]. After etching the mask-covered copper layer, the tracks remain, and the rough fiberglass base layer is exposed again.

(iii) Rigid PCB and copper tracks covered with solder resist (Figure 1c). By completely covering the etched tracks with solder resist, the track and the base substrate layer become smooth. After casting, the PDMS allows for strong permanent bonding with glass. A downside is the smoothing out of the rectangular cross section of the exposed copper tracks, resulting in a less favorable, inaccurate, and less predictable result. The geometry of said tracks will become considerably less rectangular (rounded, wider, and lower) than intended.

(iv) Flexible PCB with exposed gold-plated copper tracks (Figure 1d). After etching, the gold-plated copper tracks remain on top of a polyimide core layer. This results in sharp, rectangular, and accurate track geometry.

(v) Flexible PCB with copper tracks covered with Kapton coverlay (Figure 1e). After etching the copper tracks, a polyimide layer fitted with a pressure-sensitive adhesive, typically Kapton (DuPont, Wilmington, DE, USA), is applied to certain areas, which operate as solder resist layer. This layer, called a coverlay, smooths out the track geometry.

### 2.2. Benchmark Design

To accommodate all previously presented configurations and design elements, benchmark designs were developed that contain several configurations and design elements per PCB type to be tested. Several iterations of each type of PCB were tested; only the design of the final iteration is presented in this work. The designs can be found in Figure 2, and the design files can be downloaded in the electronic Appendix A (ESI). An overview of the properties of both types of PCBs is shown in Appendix A. To compare the quality of each PCB type, several features were incorporated in the benchmark design such as serpentine tracks, line spacing, line width, areas for surface quality testing, and droplet generators. The features are displayed in Appendix A.

### 2.3. MF Chip Making

#### 2.3.1. Design Specifications: FR-4 versus Flexible PCBs

Ordering from large-scale PCB manufacturing companies results in higher accuracy, consistency, and reproducibility compared to in-house PCB prototyping. These manufacturers have enjoyed a huge boom in popularity, making it faster and cheaper to produce small quantities of prototype PCBs and flexible PCBs [23]. Advertised prices are often less than USD5 for bespoke boards. When ordering, many options are available. Choosing non-standard options increases the price and lead time. An overview of standard manufacturing options for rigid and flexible PCBs is shown in Appendix A. In Figure 2 (PCBWay, Shenzhen, China), images of the manufactured rigid and flexible benchmark PCB are shown. Several PCBs were ordered from PCBWay.com at a price of USD 0.49 per double-sided FR-4 board for 10 pieces, which had an area of 100 × 100 mm. Flexible PCBs were ordered at a price of USD 8.78 per one-sided board for 12 pieces with an area of 75 × 100 mm. A detailed cost comparison is provided later in this publication.

#### 2.3.2. PDMS Chip Processing

Creating a functional PDMS chip involves a series of steps. This entire procedure is commonly referred to as soft lithography (SL), and it represents a fundamental approach in microfabrication for generating microfluidic devices and structures. Typically, the initial phase entails the fabrication of a master mold through lithography techniques. Subsequently, the master mold is filled with polydimethylsiloxane (PDMS), and this PDMS layer is then securely bonded to a glass substrate using plasma surface activation. In the context of soft lithography, the master mold’s intricate pattern is transferred onto the PDMS material to create a replica of the desired microscale features. The PDMS is chosen for its advantageous properties, including flexibility, biocompatibility, and ease of manipulation, making it an ideal candidate for microfluidic applications.

For a comprehensive understanding of the soft lithography process, a detailed overview is available in the supplement and visualized in Appendix A, providing insight into the various stages involved in creating functional PDMS microchips.

In this paper, a mostly identical process is used for PDMS chip making, but the silicon lithography mold is replaced by the previously described PCBs molds.

Following the bonding of the PDMS chip with glass using plasma treatment, it is imperative to activate the internal channels. To ensure cost-effectiveness and accessibility, we utilize Rain-X, a hydrophobic spray traditionally employed for rendering car windshields water-repellent, to activate the internal surfaces of the MF chip.

##### Mold Preparation Rigid PCB

To prepare the FR-4 board for casting, the sides are simply wrapped in painter’s tape to create a border for the uncured PDMS around the PCB. (Appendix A) The pressure-sensitive adhesive from the tape must be pressed firmly against the side of the PCB to ensure a leak-free result. Some adhesives interfere with the polymerization process of the PDMS, meaning the PDMS will not harden at these boundaries and remain unset and sticky; in our work, classic white masking tape (tesa^®^ BASIC Masking tape, Tesa SE, Norderstedt, Germany), was used which does not interfere with the hardening of the silicone.

##### Mold Preparation Flexible PCB

Making the mold with the flexible PCB is more complicated. Since it is not rigid, the sides cannot be easily covered with painter’s tape to create a barrier for the uncured PDMS. It also does not provide a flat and even surface, leading to uneven casting and air pockets when plasma bonding with glass. Because of local deformations in the flexible board, these distortions cannot be stretched out.

To achieve a level substrate layer, without any deformations, two methods are proposed: (i) The flexible PCB can be held against a flat surface by maintaining a strong vacuum during casting and curing (Appendix A). A bespoke 3D-printed setup was created to hold the flexible PCB on a flat surface connected to a vacuum pump. (ii) A double-sided pressure adhesive can hold down the flexible board on a rigid surface during casting, then a painter’s tape border is applied like the rigid PCB’s casting approach. (Appendix A).

Both methods worked well, but any features on the backside of the double-sided flexible PCB were visible in the cast due to the strong vacuum pulling the film flat over them. It is therefore advised to only use single-sided flexible PCBs. The back can be completely covered in coverlay to stiffen the board. The front of the board can have a copper border to increase stiffness. The second method proved to be more convenient. Following these steps will result in adequate flatness in the PDMS cast.

### 2.4. Cross Sectioning

The PCBs were processed to better visualize the track geometry as 2D inspections proved to be insufficient to get accurate insights into this. The boards were cut or sawn in pieces and were placed in custom 3D-printed containers. (Appendix A) The parts were printed from PLA (Polylactic acid) plastic filament on a fused deposition modeling (FDM) printer from Prusa Research (Prusa Mini, Prusa Research, Prague, Czechia). Then a clear two-part epoxy (Resion, Moordrecht, The Netherlands) was cast to fill the cavity of the container, encasing the pieces of circuit boards. When the epoxy completely set, the piece was clamped in a machine vice, and it was milled down with a flat-end mill up to the depth of the desired features to inspect. Finally, the top surface was sanded with increasingly fine sandpaper (from 80 to 3000 grit) (Appendix A) and then buffed with a microfiber cloth and buffing compound to make the top surface smooth and perfectly transparent, exposing a clear cross section. The PCB design included track lines on the side of the PCB so it could be easily cut and placed in the 3D-printed container with the track facing the top perpendicularly.

### 2.5. Peel Testing

A major drawback of previously published methods is the rough surface finish of the PDMS, cast on the base material of the PCB, resulting in an inadequate bonding strength between the PDMS casting and a glass microscope slide substrate [19]. To test the bonding strength of the PDMS cast on the newly proposed materials, a peel test was conducted. The adhesion of five substrate surfaces was tested; bare FR-4 substrate, flexible PCB base substrate, glass microscope slide, solder resist-covered PCB, and Kapton coverlay, each of directly imparted different surface profiles on the cast PDMS.

It was therefore important to investigate each material for suitability for long-term bonding. The existing literature on peel testing for PDMS components, including (lap) shear strength measurement, static/dynamic leakage test, and burst test [24], are similar in intent but examine different phenomena, none of which can adequately simulate the mechanical peeling of PDMS from a glass slide. Therefore, a new technique was proposed which is an adaptation of a conventional 90° peel test used for adhesive tapes.

The preparation of the 10 mm wide test samples is described in full detail in the supplement in Appendix A. The samples are mounted on a custom 3D-printed fixture to hold them in a universal testing machine to perform the peel testing (Figure 3) (MTC 100, IDM Test, Donostia-San Sebastian, Gipuzkoa, Spain).

### 2.6. Biocompatibility

It is of great value to test the biocompatibility of new MF fabrication techniques. While there are many publications that demonstrate the biocompatibility of PDMS [25,26] and the suitability of gold [27] for a range of common cell types [28,29], the cytotoxic response of polyimide materials due to the leaching of some compounds from the mold, including Kapton (DuPont, WILM, Wilmington, Delaware, USA), has the potential to contaminate the PDMS cast [30].

Here, we propose the use of flexible PCBs as a microstructuring method for fluidic chips in PDMS material, so the potential transference of material interference from the base materials, for example, the exposed gold-tinned copper tracks and polyimide base substrate, to the cast PDMS material should be tested.

Three different samples of PDMS were tested for cell viability. These 3 mm thick PDMS samples were cast on different surfaces: (i) the gold-plated copper surface, (ii) the exposed polyimide base substrate of the flexible PCB, and a section of rectangular serpentine tracks. A biopsy punch of 6 mm diameter was used to create eight duplicate material samples of the PDMS cast on the gold-plated copper, eight of the polyimide base, and sixteen samples of the serpentine tracks (50/50 mix of both conditions) (Appendix A).

These samples were cleaned from debris post punching and, after autoclaving, added to the first four columns of a standard 96-cell assay plate (per column; eight serpentine, eight gold-plated copper, eight more mixed conditions, and eight polyimide base samples). The first column of the serpentine samples was treated with a hydrophobic surfactant (Rain-X Original Rain Repellent, ITW Global Brands, Houston, Texas, USA) by spraying the surface of the sample (Appendix A). After drying, all samples were autoclaved and then placed on the bottom of the wells on a 96-well plate (Item 650209, Greiner Bio-One, Kremsmünster, Austria) in a laminar flow hood.

The choice of protocol and cell type is dictated by the intended application. For the experiment, we used the human retinal pigment epithelial cell line ARPE-19 (ATCC, CRL-2302). A detailed overview of how these cells were cultivated can be found in a publication by Boeren et al. [31]. To allow for an easier qualitative assessment of cell growth and visualization of the cells with fluorescence microscopy, the cells were modified before the experiment by transfection to express GFP, with the lentiviral construct pCHMWS-eGFP-Ires-Hygro selected for the expression of GFP using hygromycin and sorted to obtain a homogenous GFP expression using fluorescence-activated cell sorting.

To allow for quantitative analysis, we chose to perform a resazurin-based assay to quantify the number of viable cells that were grown during the seven-day period of cultivation. The protocol is based on the reduction of the oxidized non-fluorescent blue resazurin to a red fluorescent dye (resorufin) by the mitochondrial respiratory chain in live cells. As such, the amount of resorufin measured was directly proportional to the number of living cells. During incubation, the medium was changed daily. After a one-week (7-day) incubation period, resazurin was added without removing the medium. Living cells can reduce resazurin to resorufin, which is pink in color. An increase in resorufin, as measured with a spectrophotometer, allows for the quantification and comparison of cell growth and viability in different conditions.

## 3. Results

### 3.1. Cross-Sectional PCB Feature Analysis

#### 3.1.1. Track Geometry

In the context of FR-4 PCB production, our examination highlights a trapezoidal cross-sectional track profile, where the etching process, specifically isotropic etching, plays a critical role. This method, unlike anisotropic etching that etches at different speeds depending on crystal orientation, proceeds uniformly across all orientations, leading to the observed concave sides of the tracks. This uniform etching contributes to the characteristic concave appearance, further emphasized by tin plating on the tracks to prevent oxidation (Figure 4a).

To enhance the adhesion between the cast and glass, a flat surface surrounding the tracks is essential. However, the application of solder resist on the PCB deliberately leaves a gap around the tracks, with an offset of 2–3 mil (50–75 µm) to avoid covering them, inadvertently creating a geometry that is not conducive to optimal adhesion. This configuration, with a pronounced gap between the solder resist and the tracks, leads to the formation of protruding features post casting, which disrupts the ability to achieve a flush bond with a flat surface. The significance of this gap and its impact on molding suitability are illustrated in Figure 4a.

In contrast, the adjacent track on the right is entirely coated with solder resist (as shown in Figure 4b). The additional height added by this layer on top of the track is considerably thinner compared to the height increment alongside the track. This differential in thickness contributes to the rounding of the track. As a result, the overall height of the structure deviates significantly from the intended height, assuming a broader and more rounded form, thereby yielding unpredictable outcomes when utilized as a mold. This phenomenon is consistent across both 1oz and 2oz thickness of the copper track (1 and 2 oz weight) configurations, which is depicted in Figure 4c,d.

The cross-sectional shape of the tracks on the flexible PCB is more rectangular in shape compared to the FR-4 PCB. However, it still exhibits similar concave shaping of the side walls due to the isotropic wet etching that is used during production. The gold plating forms an almost negligible thin layer of around 3 µm which is significantly thinner compared to the tinning added to the FR-4 PCB. In Figure 4e,f, the cross section shows the difference between the covered and non-covered flexible PCB.

Figure 4f shows the flexible PCB tracks covered in cover lay, resulting in very wide smoothing of the tracks, rendering them completely unusable for molding.

#### 3.1.2. Track Spacing

The minimum manufacturable track width and track spacing for the FR-4 PCB specified by the manufacturer (PCB Way, Shenzhen, China) is 6 mil or 152.4 µm. More accurate production is possible down to 3 mil or 76.20 µm but significantly increases the cost of the PCB multiple times compared to fabrications using the 6 mil minimum specifications.

For the flexible PCB, the standard minimum track width and track spacing is 60 µm. Although a spacing of 60 µm is possible on the flexible PCB and does produce usable PDMS chips, a minimum of 100 µm is recommended to ensure the area between the tracks is etched all the way to the base substrate (Figure 4g,h).

The thicker the copper foil layer, the more spacing is required. The minimum spacing of 100 µm is suitable for the 1 oz copper weight. When the copper has a 2 oz weight, the etching must go deeper, and thus a minimum spacing of 8 mil (one-thousandth of an inch) or 203 µm is required. Therefore, a 1oz copper thickness was selected as most suitable for the flexible PCB molds.

#### 3.1.3. Track Width

Our benchmark boards demonstrate the capability to accommodate a wide range of track widths, spanning from 60 µm to 2000 µm. To ensure accuracy, measurements across various points were performed to compare the intended design widths. The track accuracy of the rough FR-4 boards is not assessed here, owing to the inadequate bonding of the FR-4 surface with glass—a detail which is further elaborated upon in Section 2, under Peel Testing.

The application of a solder resist layer on the FR-4 boards yields a smooth, rounded track geometry, complicating the precise delineation of track edges when observed from above through conventional optical microscopy. This complexity introduces an unpredictable dimension offset, rendering practical measurements challenging. The variance in track width could markedly surpass the original design specifications. In contrast, we were able to ascertain accurate track geometry measurements for the flexible PCB benchmarks.

Illustrated in Appendix A, the image showcases three interconnected serpentine tracks, maintaining a uniform spacing of 100 µm between them. Track widths are depicted to vary across different figures: 250 µm in Appendix A, 100 µm in Appendix A, and 60 µm in Appendix A, with extensive measurements conducted for each specified width. These assessments underscore a commendable replication of the intended designs, with average measurements closely aligning at 247 µm, 101 µm, and 59 µm for the designated widths, respectively. The calculated average relative offset fluctuates between 1.5% and 3.4% across all tracks, as detailed in Appendix A. Further, detailed close-ups and scanning electron microscope images presented in Appendix A highlight the precision and definition achieved in the track geometries.

### 3.2. Surface Roughness

In this investigation, we explore the critical role of surface roughness in the mold-making process, highlighting its impact on the efficiency of polydimethylsiloxane (PDMS) bonding to glass and its significance in fluid dynamics within microfluidic channels. Our analysis encompasses a range of surfaces, documented through the imagery in Figure 5. This matrix, divided into two primary sections, offers an extensive visual overview of the surfaces examined.

The top section of Figure 5(a1–d3) details the surface features of PDMS channels as cast on tinned copper tracks (Figure 5(a1–a3)), solder resist-covered tracks (Figure 5(b1–b3)), and gold-plated copper tracks with widths of 60 µm (Figure 5(c1–c3)) and 200 µm (Figure 5(d1–d3)). Arranged in a 3 × 4 matrix, these images present a clear view at both 125 µm and 50 µm image scales, which reveal detail on surface textures and potential irregularities transferred during molding.

The bottom section of Figure 5e–j reveals a 2 × 3 matrix that highlights the FR-4 (Figure 5e,h), solder resist (Figure 5f,i), and polyimide (PI) (Figure 5g,j) surfaces. This arrangement transitions from base surface images to a 3× digital zoom perspective, offering insight into the subtle differences in roughness and texture across various materials and processing methods.


**Bonding Surface Analysis:**
FR-4 Substrate: The consistent pitting observed across FR-4 substrate molds is vividly captured in images e and h. Research reports roughness of around 1 µm Ra [32].Solder Resist: Micron-scale bubbling, attributable to air entrapment within the solder resist substrate surfaces, is depicted in Figure 5f,i.Polyimide (PI): The PI substrate molds, exhibiting uniform surface roughness, are showcased in Figure 5g,j. Research reports suggest significantly better roughness of around 250 nm Ra [32].



**Channel Surface Roughness:**
Tinned Copper: Minor pitting and more pronounced scratches, varying in orientation on the cast surfaces of tinned copper substrates, are detailed in Figure 5(a1–a3).Solder Resist-Coated Channels: The roughness characteristics akin to the solder resist bonding surface are documented in Figure 5(b1–b3).Gold: The gold substrate casts, featuring less frequent pitting and unidirectional scratches, are visualized in Figure 5(c1–c3) and Figure 5(d1–d3), with a focus on their orientation relative to channel directions. The roughness of the coper used for PCB making has a roughness of <0.5 µm Ra [33].


This nuanced approach to both visual and textual documentation highlights the specific and process-induced variations in surface textures, emphasizing the complexities involved in achieving optimal bonding and fluid control within microfluidic systems.

Surface roughness is crucial for effective glass adhesion and fluid dynamics in microfluidic devices. Molds created through conventional soft lithography typically exhibit surface roughness within a low nanometer range. However, our fabrication technique results in surfaces with roughness in a low micrometer range. The appropriateness of this roughness level largely depends on the specific application. Given our method’s capability to achieve features down to approximately 60 µm, the roughness observed is not anticipated to adversely affect fluid dynamics significantly.

### 3.3. Peel Testing

In our study, peel testing was meticulously performed to evaluate the adhesion between PDMS casts and glass surfaces, following a detailed protocol outlined in the methods section of our supplement. Examples of the surface roughness of some of these interfaces are shown in Figure 5e–j. This testing involved PDMS casts that were formed using various substrates as molds. The results, presented in Appendix A and summarized in Table 1, illustrate the bonding strengths achieved when these PDMS casts were subsequently bonded to glass via plasma treatment. As a benchmark, we utilized a PDMS cast created from the surface of a glass microscope slide, anticipating this to provide the highest level of bonding strength due to its smooth surface. This set the standard against which we compared the bonding capabilities of PDMS casts formed from other substrates: the FR-4 core, flexible PCB core, Kapton tape, and solder resist.

The outcomes highlighted in Appendix A and Table 1 underscore the significant influence of the substrate material used to form the PDMS casts on the final bonding strength with glass. PDMS casts formed on Kapton and solder resist surfaces demonstrated exceptionally strong bonds to glass, surpassing the PDMS’s tear strength, with average maximum forces of 2.55 N for Kapton and 3.20 N for solder resist, respectively. In these cases, the PDMS material tore rather than peeled from the glass, indicating a bond stronger than the PDMS itself. In contrast, PDMS casts using the FR-4 core as a mold showed minimal or no adhesion to glass. In contrast, casts formed using the flexible PCB core as a mold bonded to glass with an average maximum force of 2.61 N, indicating a robust adhesion nearly on par with the glass-to-glass baseline.

Interestingly, variability was observed in the performance of PDMS casts replicated from glass substrates; two out of five samples experienced tearing, and the remaining samples showed less effective adhesion compared to those formed on Kapton and solder resist surfaces. This highlights the critical role that the initial substrate material, used only to shape the PDMS casts, plays in determining the final adhesive strength between the PDMS and glass, despite all tests being conducted exclusively between PDMS casts and glass surfaces.

The results from our peel testing offer compelling quantitative evidence on the adhesion performance between PDMS casts and glass, highlighting the variability based on the substrate used for casting. Specifically, PDMS casts formed on the FR-4 substrate exhibit notably poor bonding strength when adhered to glass, reinforcing the consensus in existing research [10,18] that the FR-4 surface is ill suited for direct PDMS casting without prior modification.

Conversely, our testing reveals that PDMS casts created using the flexible PCB as a mold significantly outperform those formed on the FR-4 substrate in terms of bonding strength to glass. The average maximum bonding strength observed for PDMS cast from the flexible PCB substrate reached 2.61 N over a width of 10 mm, equivalent to 13.05 kN/m^2^. This distinct difference underscores the superior compatibility of the flexible PCB surface for PDMS casting, especially when the goal is to achieve robust adhesion between the PDMS cast and glass surfaces. Both geometry and roughness contribute to the final bonding strength of the PDMS cast with glass. A comprehensive overview of these considerations is provided in Table 2.

### 3.4. Cost Comparison

Lithography is renowned for its precision in soft lithography for microfluidic applications but comes at a significantly higher cost compared to rapid prototyping methods like 3D printing and PCB fabrication. This study highlights that PCB fabrication, with rigid FR-4 PCBs (covered in solder resist) priced at USD 0.49 and flexible PCBs at USD 8.87, offers a cost-effective alternative for rapid prototyping with a good balance of feature resolution and fabrication speed. While lithography provides unmatched accuracy, especially using quartz masks for sub-micron features, the PCB method is advantageous for its lower cost and flexibility in design iterations similar to other conventional prototyping methods [34,35]. A comprehensive cost comparison is provided in Table 3. The comparison underscores the importance of selecting a fabrication method that aligns with the specific needs of the microfluidic device, considering both the required precision and budget constraints. Stereolithography (SLA) 3D printing is added to the comparison since it is considered one of the most accurate additive manufacturing methods and shows great potential for high-detail mold making. Several elements impact the cost of microfluidic device fabrication, including design intricacy, manufacturing speed, and the number of units ordered. Lithography’s use of circular wafers contrasts with PCBs’ rectangular format, which is easily linkable to the dimensions of microscope slides. A detailed comparison in Table 3 aids in method selection, with costs derived from various quotes and data on UV mask products. For price inquiries, a standard flexible PCB layout was referenced. Lithography expenses encompass mask, SU-8 mold, and wafer cutting costs, outlined as USD 30, USD 420, and USD 100, respectively.

A detailed comparison across five microfabrication techniques, assessing factors such as precision, cost, and production timeline, is presented in Table 3. This analysis spans from traditional FR-4 PCB to advanced lithography and SLA printing, highlighting each method’s strengths and limitations in terms of feature size, material cost, and application suitability. This comparison is instrumental for selecting the most appropriate fabrication approach tailored to specific microfluidic device requirements, optimizing for efficiency and cost-effectiveness.

### 3.5. MF Devices—Benchmark Testing

#### 3.5.1. Serpentine Channel and Color Mixer

Holding and transporting fluids are essential for the working of any MF chip. Both 1 and 2 oz rigid PCBs and 1 oz flexible PCBs were put to the test. An overview of several PDMS chips is presented in Appendix A.

The smallest possible features produced without deviating from standard manufacturing specifications are tracks of 60 µm width and 60 µm spacing for the flexible PCB and 100 µm wide and 100 µm spacing for the rigid PCB. The flexible PCB shows perfect replication, resulting in water-tight and properly bonded channels with sharp outlines.

The chips made on a rigid PCB show feathering around the edges, leading to unpredictable and inaccurate channel geometries. The chip, however, is bonded very well to the glass microscope slide. Due to this reason, it can be concluded that fluid transport is possible, but internal volumes and fluid behavior are hard to control.

Color mixers were also produced with both PCB types. Due to feathered and irregular track outlines in the chips made with the FR-4 PCB, the flow inside is not sufficiently laminar, resulting in the mixing of two combined streams. The channels produced with the flexible PCB, however, show controlled laminar flow, allowing the two dyes to stay separate (Appendix A).

#### 3.5.2. Droplet Generator

Several droplet generator designs were tested such as T-junction droplet generators and flow focusing droplet generators. Droplets are generated by combining two immiscible phases at a narrow section, resulting in monodispersed droplets. In-house-constructed syringe pumps were used to actuate the flow at rates between 0.1 and 5 µL/min. Two liquids were used; for the continuous phase, mineral oil (C_16_H_10_N_2_Na_2_O_7_S_2_, Arcos Organics, Thermo Fisher Scientific, Waltham, MA, USA) was used, and for the dispersed state, a water-based dye (Winson and Newton Ink, London, UK) was used.

The smoothing of the tracks on the FR-4 PCB covered in the solder resist results in inaccurate and faded channel outlines replicated in the PDMS (Appendix A). Due to this, the liquids inside the chip bypass the intended design features, leading to unstable and unpredictable droplet generation (Appendix A).

The droplet generator made by the replication of the 1oz flexible PCB showed accurate, well-defined features, with a junction that narrows down to a size of 75 µm with a channel height of 35 µm (Figure 5a,b). Using the open-source image processing software package Fuji/ImageJ (Release 2.15.1) [45], the size distribution of a series of droplets was measured from a video of the droplet generator. The droplets were fitted as ellipses, and the equivalent droplet diameter was extracted as the mean of the lengths of the major and minor axes of the ellipse. For the cross-junction droplet generator shown in Figure 6a, 108 droplets were analyzed. This is shown in Figure 6c,d with a histogram binned with 1 µm width increments (shown in blue). Additionally, a normal distribution was fitted to these data (shown in red), revealing a mean equivalent droplet diameter of 70.18 µm and a standard deviation of 3.54 µm, keeping in mind the width of the channel was 250 µm, and the width of the narrowing in the cross-junction was 75 µm. Similar analyses on 77 droplets from the T-junction, as shown in Figure 6b, revealed a mean equivalent droplet diameter of 93.98 µm and a standard deviation of 2.73 µm. Again, the junction geometry had a channel width of 250 µm, and the width of the narrowing in the T-junction was 75 µm, with a channel height of 35 µm.

### 3.6. In Vitro Biocompatibility

Utilizing an inverted fluorescence microscope (Zeiss Axio Observer Z.1), we observed consistent growth patterns across all samples when compared to the control conditions. However, a notable exception was observed in the gold-coated copper PDMS, which showed a visually distinctive reduction in cellular growth compared to the control group, as shown in Appendix A.

The resazurin assay analysis hinges on a defined basis: % inhibition is benchmarked with medium-only conditions as the negative control (equating to 100% inhibition), while cells grown in conditions without PDMS samples serve as the positive control (representing 100% growth). The relative growth under experimental conditions is expressed compared to this positive control.

Our preliminary assessments, illustrated in Figure 7, suggest a promising direction for further investigation. In conditions treated with Rain-X, we observed a notable increase in average growth rates when compared to the control group (*p* = 0.049). This observation underscores the potential of surface treatments in enhancing cell viability yet highlights the necessity for in-depth future research to comprehensively evaluate metabolic activity and viability across a broader spectrum of conditions. This, however, indicates a favorable effect of the Rain-X coating, resulting in an average growth rate of approximately 125%, which is 25% higher than the control group. In the copper/gold condition, there was a substantial reduction in growth compared to the control (*p* = 0.0003), signifying a marked delay in growth. This suggests that material transfer from the substrate to the PDMS significantly diminishes cell growth/viability, resulting in an average growth rate of approximately 62% in comparison to the control.

Both the mixed gold/copper and Kapton PDMS conditions did not exhibit any significant differences compared to the control. Average growth rates for these conditions were approximately 97% and 96%, respectively, despite the presence of up to 50% gold/copper on the surface. This suggests that these materials had minimal impact on cell growth in comparison to the control group.

The observed increase in surface compatibility in Rain-X-coated PDMS may be attributed to its hydrophobic surface properties or its protective effects against factors that could hinder cellular compatibility. However, it is noteworthy that the slower growth seen on the copper/gold substrate is a consequence of an extended incubation period (7 days), which corresponds to a daily reduction in growth rate of approximately 8–9% compared to the control group. For many microfluidic applications, such as mixing or transport, short contact periods are typical, leading to minimal interference. Furthermore, Rain-X treatment has been proven to significantly enhance the surface quality of microfluidic chips, as demonstrated by Karade V. [46], while also establishing a biocompatible protective barrier against potential PDMS leachates.

## 4. Conclusions

In this paper, we explored an alternative approach to microfluidic chip fabrication utilizing printed circuit board (PCB) technology, which diverges from the conventional methods employed in the field, particularly soft lithography. Our analysis spanned four distinct PCB variations, each evaluated for its potential utility in creating master molds for microfluidic applications. This nuanced examination led to insights into the strengths and constraints associated with each PCB type, guiding the selection of the most suitable method based on specific device requirements, including considerations of precision, cost-efficiency, and production speed.

Our findings reveal that while traditional FR-4 PCBs exhibit poor adhesion to glass, rendering them unsuitable for direct use in microfluidic chip fabrication, the application of solder resist substantially improves their adhesion properties. This makes solder resist-coated FR-4 PCBs viable for applications that do not demand high geometric fidelity. Flexible PCBs without a coverlay demonstrated good adhesion, affirming their utility in fabricating functional microfluidic devices, as illustrated through successful implementations like droplet generators and color mixers. However, the use of a coverlay on flexible PCBs was found to hinder precise microfluidic molding due to excessive track smoothing, thus deemed unsuitable for high-precision applications.

Comparatively, while lithography stands out for its unparalleled precision within the realm of soft lithography, it is accompanied by significant cost barriers, with mold prices ranging from USD 600 to USD 1000. In stark contrast, our PCB-based approach offers a markedly more economical alternative, with mold costing between USD 0.5 and USD 10. This affordability is pivotal, enhancing the development process by providing a prototyping alternative rivaling and, in some cases, surpassing the precision of expensive rapid prototyping methods available in the field.

Notably, our method using flexible PCBs as molds enables the structuring of features down to 60 µm with channel heights beginning at 9 µm and tolerances maintained below 5%, with a maximum deviation of 6 µm. Although it does not achieve the sub-micron precision afforded by soft lithography, it presents a compelling option for many microfluidic applications that can accommodate these tolerances.

Furthermore, to enhance the accessibility and cost-effectiveness of our method, we employed a common water-repellent, Rain-X, to activate microfluidic chip internal volumes. This simple yet effective approach proved to be biocompatible and significantly enhanced chip performance without imposing a substantial cost burden. Even though untreated flexible PCBs showed reduced cell viability when tested for 7 days, shorter contact could yield acceptable results.

In conclusion, this study introduces a cost-effective, efficient alternative to traditional microfluidic device fabrication methods, thereby expanding the toolkit available to researchers and developers in the field. It provided a structured guide for selecting appropriate fabrication methods and highlighted the promising potential of PCB technology in microfluidic applications.

## Figures and Tables

**Figure 1 micromachines-15-00425-f001:**
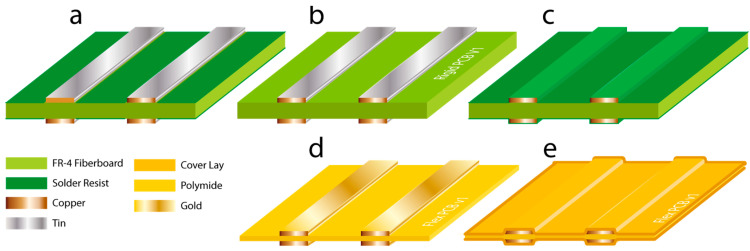
This figure provides an overview of the cross sections of different types of PCBs tested in this work. (**a**) Exposed, tinned copper tracks with solder resist. (**b**) Exposed, tinned copper tracks. (**c**) Copper tracks covered with solder resist. (**d**) Exposed, gold-plated copper tracks on flexible PCB and (**e**) copper tracks covered with coverlay on flexible PCB.

**Figure 2 micromachines-15-00425-f002:**
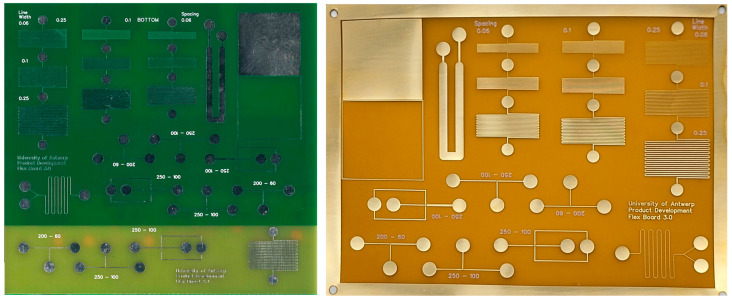
Image of the manufactured rigid FR-4 PCB (**left**) with top part covered in solder resist and bottom part bare substrate and flexible PCB (**right**) without coverlay.

**Figure 3 micromachines-15-00425-f003:**
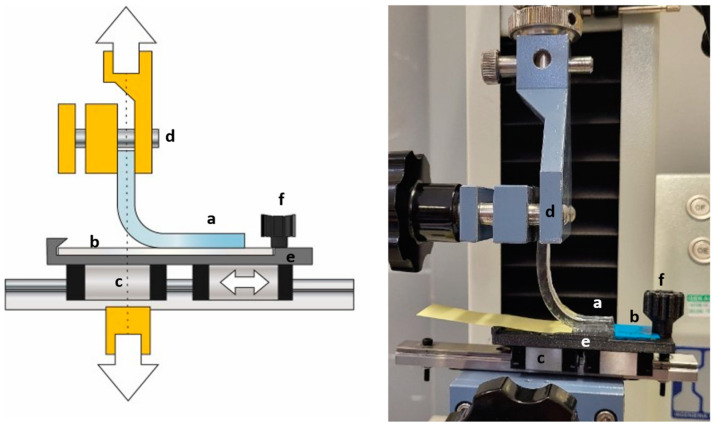
This figure provides an overview of the peel test setup: (a) bonded area of PDMS, (b) microscope slide, bonded to PDMS, (c) linear rail allows for low friction movement, ensuring a 90° pull test angle at all times, (d) top clamp, (e) 3D-printed fixture, and (f) a screw, holding the microscope slide in place.

**Figure 4 micromachines-15-00425-f004:**
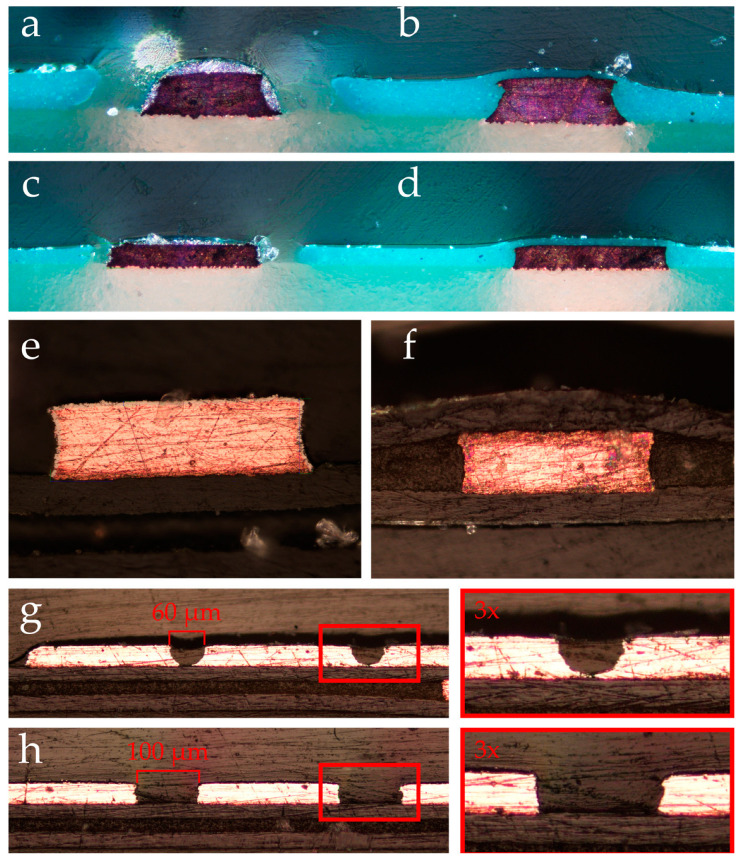
This figure provides a comprehensive cross-sectional overview of various PCB configurations: (**a**) depicts a 2 oz thick copper FR-4 PCB with a 200 µm wide tinned track featuring sides covered in solder resist. (**b**) illustrates the same PCB with the tinned track entirely covered in solder resist. (**c**) shows a 1 oz thick PCB with a 250 µm wide exposed copper track. A noticeable gap in the screen print and additional tinning on the copper track are evident. (**d**) presents the PCB from (**c**) with the exposed track fully covered in solder resist. (**e**) highlights a cross-sectional view of a bare, gold-coated track on a flexible PCB. (**f**) depicts the same flexible PCB track, but this time, it is covered by a coverlay. (**g**) displays cross-sectioned tracks on a flexible PCB with 60 µm spacing. (**h**) shows similar cross-sectioned tracks on the flexible PCB but with a 100 µm spacing. The right side of (**g**,**h**) provides a three times digital magnification for enhanced clarity.

**Figure 5 micromachines-15-00425-f005:**
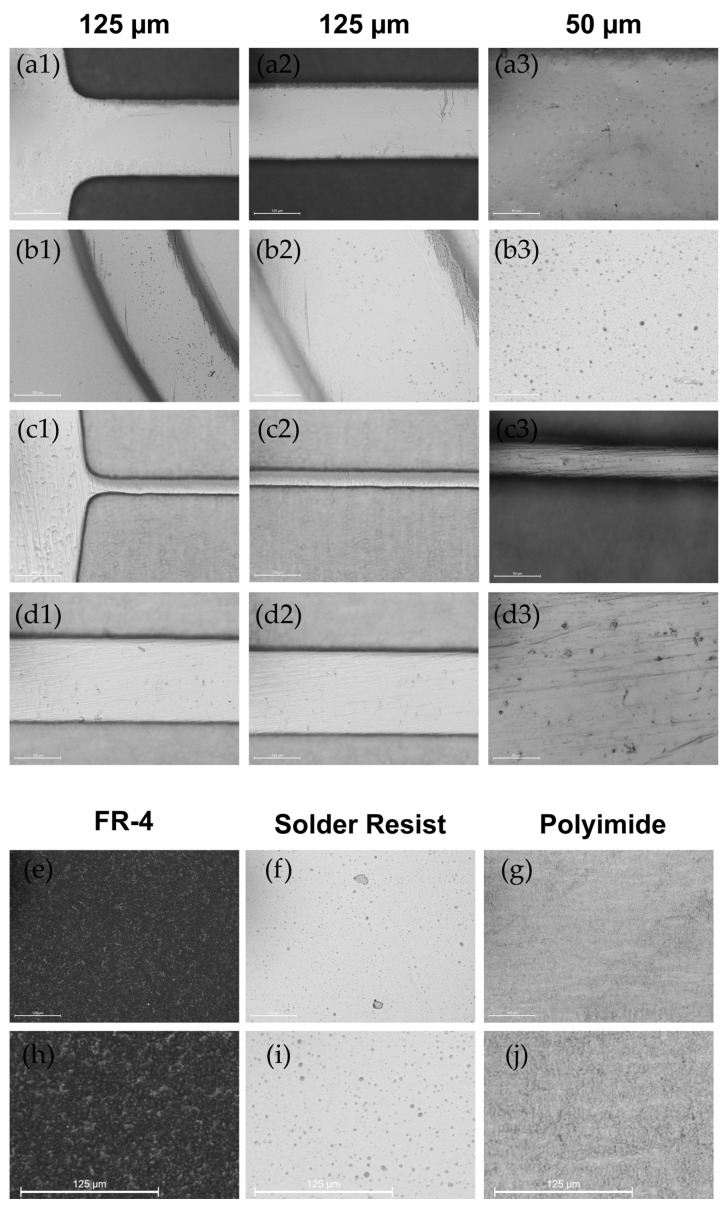
A detailed matrix illustrating a variety of substrate surfaces and their magnified specifics. The top section (**a1**–**d3**) presents a 3 × 4 matrix of PDMS channels cast on tinned copper tracks (**a1**–**a3**), solder resist-covered tracks (**b1**–**b3**), and gold-plated copper tracks at widths of 60 µm (**c1**–**c3**) and 200 µm (**d1**–**d3**), at scales of 125 µm and 50 µm to detail texture and molding irregularities. The bottom section features a 2 × 3 matrix, highlighting the FR-4 (**e**,**h**), solder resist (**f**,**i**), and PI (**g**,**j**) surfaces, with the top row offering base surface images and the bottom row showcasing a 3× digital zoom. This elaborate arrangement facilitates an in-depth comparison of surface roughness and texture across different materials and magnifications, serving as a critical visual aid in analyzing bonding efficacy and fluid dynamics in microfluidic applications.

**Figure 6 micromachines-15-00425-f006:**
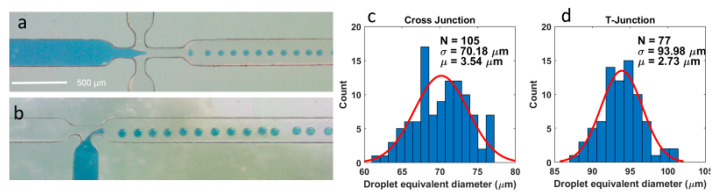
(**a**) This figure showcases a working cross-junction droplet generator made with a flexible PCB. (**b**) Working T-junction droplet generator made with a flexible PCB. (**c**) Histogram showing the monodispersity of droplet generated by the cross-junction. (**d**) Histogram showing the monodispersity of droplet generated by the cross-junction droplet generator.

**Figure 7 micromachines-15-00425-f007:**
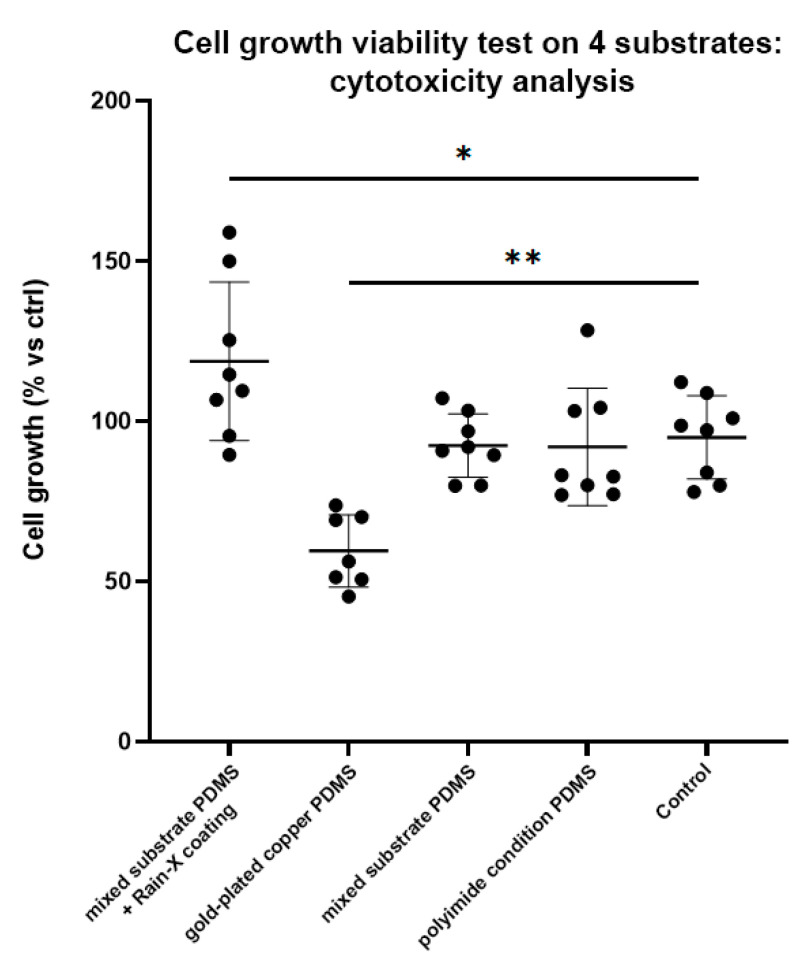
This figure provides a look at the cell growth viability on different substrates. Cell growth (% vs ctrl) on the y-axis, substrates on the x-axis. Lines with whiskers indicate the median with interquartile range from the three independent experiments combined. The *p* values reported were obtained from a linear mixed model that was adjusted for multiple testing using Tukey HSD all-pairwise comparison test. * *p* < 0.05, ** *p* < 0.01.

**Table 1 micromachines-15-00425-t001:** 90° peel test of PDMS on glass slide, measured adhesion force (N) ^1^.

	Sample 1	Sample 2	Sample 3	Sample 4	Sample 5	Sample 6	Max.	Avg.	Std.
FR-4	0.42	0.3	0.21	0.42			0.42	0.33	0.1
Flex	2.3	2.15	3.4	2.93	2.51	2.35	3.4	2.61	0.47
Glass	4.03	3.77	2.59	2.26	1.76		4.03	2.88	0.98
Kapton	2.9	2.7	2.67	2.68	1.76	2.59	2.9	2.55	0.4
Solder Resist	3	3.67	2.97	3.1	2.87	3.59	3.67	3.2	0.34

^1^ The peel test was conducted using strips with a width of 10 mm. To obtain the adhesive strength values in newtons per millimeter (N/mm), the measured force should be divided by ten.

**Table 2 micromachines-15-00425-t002:** This table compares various materials and methods used for microfluidic chip fabrication, focusing on their bonding ability, channel/track features, and additional relevant notes. It highlights the challenges and advantages of using FR-4, FR-4 with solder resist, flexible PCBs, and lithography in the development of microfluidic devices, underscoring the impact of material choice on device fabrication and functionality.

**Material/Method**	**Bonding Ability**	**Channel/Track Features**	**Additional Notes**
FR-4	Not able to bond	Rough substrate	Substrate too rough for effective bonding
FR-4 + solder resist	Impossible due to gaps	Rounded effect from tinning; unreliable shape, wide channels	Good for fluid transport and millifluidics
FR-4 solder resist only	Good adhesion	Unreliable shape, wide channels	Suitable for fluid transport, offers better adhesion
Flex PCB	Better/sufficient adhesion	Accurate, up to 60 µm width, close spacing, good surface finish	Limited biocompatibility, gold-coated (1 U thickness)
Flex PCB + coverlay	Not tested	Too rounded for use	-
Lithography	Excellent	High precision, suitable for features smaller than Flex PCB	Ideal for applications requiring small features or smooth surfaces

**Table 3 micromachines-15-00425-t003:** Overview of five microfabrication methods: FR-4 PCB, flexible PCB, lithography 1 with a plastic mask (Class 3, 256k dpi), lithography 2 with a chrome mask on quartz (Class 4, 512k dpi), and SLA printing. It evaluates each method based on tolerance, minimum feature size, cost, height range and steps, surface roughness, aspect ratio, surface size, area, cost per surface area, and production time. This comparison provides a concise overview to aid in selecting the most suitable fabrication method for specific microfluidic device requirements, considering factors like precision, cost-efficiency, and production speed [36,37,38,39,40,41,42,43,44].

	FR-4 PCB	Flexible PCB	Lithography 1	Lithography 2	SLA Printing
Tolerance	Very low	<6 µm	<0.4 µm	<0.2 µm	200 µm
Min. feature size	100 µm	60 µm	2 µm	1 µm	200–300 µm
Feature height	35–105 µm	9 µm–88 µm	1.5–100+ µm	1.5–100+ µm	32–1000+ µm
Height accuracy	35 µm	9 µm	<1 µm	<1 µm	32 µm
Aspect ratio	1:2	12:1	1:1–5:1	>10:1	>10:1
Bonding surface roughness ^1^	1 µm Ra	250 nm Ra	10 nm Ra	10 nm Ra	3–10 µm Ra
Channel surface roughness	>±10 µm Ra	>±5 µm Ra	10 nm Ra	10 nm Ra	3–10 µm Ra
Surface shape	100 × 100 mm (2×)	75 × 100 mm	⌀ 152.4 mm	⌀ 152.4 mm	100 × 100 mm
Surface area	200 cm^2^	75 cm^2^	182 cm^2^	182 cm^2^	100 cm^2^
Cost ^2^	USD 0.5	USD 10	USD 600	USD 1110	USD 5
Cost/surface area	0.003 $/cm^2^	0.133 $/cm^2^	3.296 $/cm^2^	6.098 $/cm^2^	0.05 $/cm^2^
Production time	1–4 days	6–7 days	7–10 days	7–10 days	4–6 days

^1^ Roughness values of FR-4 PCB and flexible PCB are estimates based on a measuring a large sample of random artifacts, visible on the surface. This measurement is merely an indication of magnitude for comparison purposes. ^2^ Cost estimates can be found the electronic supplement.

## Data Availability

The original contributions presented in the study are included in the article, further inquiries can be directed to the corresponding author.

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
