# Peer review of "Achieving High-Precision, Low-Cost Microfluidic Chip Fabrication with Flexible PCB Technology"

_micromachines, 2024, doi:10.3390/mi15040425_

Round 1
Reviewer 1 Report
Comments and Suggestions for Authors
Two types of PCBs were used as alternative master molds for soft lithography in this work, and in-depth explorations of using PCBs as a mold substrate were conducted. Although the manuscript describes all the experiments in detail, the novelty of using flexible PCB as an alternative master mold is poor, since this approach comes to mind easily for the researchers in the field of microfluidics. And the scientific issues contained this approach are not discussed. In addition, tht quality of English language should be further improved. There are some low-level English writing errors, for example, “a novel improved method” was miswritten as “n novel improved method” on line 80, the first letter of comprehensive should be capitalized in Fig4. After evaluating this work comprehensively, I would not recommend to accept this manuscript.
Comments on the Quality of English LanguageTwo types of PCBs were used as alternative master molds for soft lithography in this work, and in-depth explorations of using PCBs as a mold substrate were conducted. Although the manuscript describes all the experiments in detail, the novelty of using flexible PCB as an alternative master mold is poor, since this approach comes to mind easily for the researchers in the field of microfluidics. And the scientific issues contained this approach are not discussed. In addition, tht quality of English language should be further improved. There are some low-level English writing errors, for example, “a novel improved method” was miswritten as “n novel improved method” on line 80, the first letter of comprehensive should be capitalized in Fig4. After evaluating this work comprehensively, I would not recommend to accept this manuscript
Reviewer 2 Report
Comments and Suggestions for Authors
The authors presented the mold fabrication approach in PDMS soft-lithography using flexible PCB technology. Although the approach is interesting I have some comments to improve the manuscript further.
(1) Paragraph 2 in Introduction (soft lithography vs PCB): it is not a good comparison between the conventional soft lithography vs PCB approach. The authors should compare both approaches strictly in mold fabrication. E.g.
(a) Line 42: Fabricating the master molds requires cleanroom facilities. PCB fabrication needs cleanroom facilities as well.
(b) Line 43: Ordering can have lead times of 6 to 8 weeks. This strictly depends on the company's turnaround time, from design to prototype (not only mold fabrication). I can make a mold in 3 hours.
(c) Line 45/46: The cost is expensive up to USD 15,000. Again, this is not a realistic comparison and the report is in 2014. You can easily find a cheaper option in China. The cost was high at that time because it was still a new technology, there were limited fabrication foundries, there were less comprehensive guidelines in design, etc. I do not agree it applies to the present day.
(d) Line 49: The equipment ranges $20,000 to $50,000. How about the equipment for PCB fabrication?
The authors should provide a direct comparison of fabricating the mold using soft lithography vs PCB approach, such as the material cost, but not based on foundry or service provider or based on ordering PCB from a foundry vs setting up a soft lithography cleanroom.
(2) Line 158: Please provide the total cost of each PCB based on the final design.
(3) Paragraph layout: The manuscript has quite several paragraphs with a single sentence. Please restructure the whole manuscript.
(4) Line 179-180: Again, you can order a PDMS mold to avoid needing a cleanroom.
(5) Figures 4e and f: What is the cause of the concave sidewall?
(6) What is the range of heights and aspect ratio (height/width) of microchannels that can be fabricated using the five mentioned PCB approaches? As compared to conventional soft lithography.
(7) Peel testing: I am confused with the peel testing. The peel testing should be conducted with the PDMS casts from the five PCB approaches against a glass slide. However, in Lines 235-237, the peel test was conducted with PDMS cast (From which approach?) against five different surfaces. Table 1 suggests the same (Do you intend to use the PDMS with PCB board? But this is not the focus of the manuscript). But Line 380: FR-4 not bonded to the glass (Do you mean you bond FR-4 on glass?). Line 386: the bonding strength between the glass and PDMS cast from the FR-4 is poor (here is the PDMS cast and glass). Please restructure this part and focus on the peeling test between PDMS casts from each PCB approach + conventional soft lithography against the glass slide for comparison.
(8) Surface roughness: Please show zoom-in images of microchannels fabricated by each approach and different channel widths and discuss the quality of the microchannel. Does the PCB approach provide roughness comparable to conventional Soft lithography using a plastic mask or chrome mask?
(9) Please provide the cross-sectional dimension of the chips being used in Figure 5.
(10) Discussion and conclusion: Please discuss and compare the new PCB method vs conventional soft lithography in terms of material cost, smallest structures, aspect ratio, roughness, bonding strength, etc., and highlight the advantages and limitations of the PCB approach and its potential application areas. For instance, it is not suitable when a small feature size or small surface roughness is desired.
(11) ESI Figure S11: The descriptive texts highlighted Figures S9 instead. Please correct the typos.
(12) Please proofread the whole manuscript for typos, e.g., Line 40: 500 nm, Line 46: 15,000 or 15.000, Line 80: a novel improved method, Line 339: 60 um
Round 2
Reviewer 1 Report
Comments and Suggestions for Authors
All the comments are addressed, the manuscript can be accepted in present form.
Reviewer 2 Report
Comments and Suggestions for Authors
The authors have revised the manuscript based on the reviewers' comments. It is suitable for publication with the following minor comments.
(1) Line 54 - 57: Please merge the sentences into a single paragraph.
(2) Line 64: miss out the '.' at the end of the sentence.
(3) Line 166: Should be Table 3 but not Table 2.
(4) Line 461 to 465: Merge to a single paragraph.
(5) Line 495: Not a complete sentence.
(6) Table 3 Feature heights of PCB: Please include a statement in the manuscript to discuss how different heights can be achieved.
(7) Table 3 Aspect ratio of lithography: Typically should be within 10:1 but not 5:1 (DOI 10.1088/1361-6439/ac00c8)
